# Adaptive Predefined-Time Sliding Mode Control for QUADROTOR Formation with Obstacle and Inter-Quadrotor Avoidance

**DOI:** 10.3390/s23052392

**Published:** 2023-02-21

**Authors:** Hao Liu, Haiyan Tu, Shan Huang, Xiujuan Zheng

**Affiliations:** Key Laboratory of Information and Automation Technology of Sichuan Province, Department of Automation, College of Electrical Engineering, Sichuan University, Chengdu 610065, China

**Keywords:** quadrotor formation, artificial potential field method, virtual force, RBF neural network, predefined-time sliding mode control

## Abstract

In this paper, aiming at the problem of control and obstacle avoidance in quadrotor formation when mathematical modeling is not accurate, the artificial potential field method with virtual force is used to plan the obstacle avoidance path of quadrotor formation to solve the problem that the artificial potential field method may fall into local optimal. The adaptive predefined-time sliding mode control algorithm based on RBF neural networks enables the quadrotor formation to track the planned trajectory in a predetermined time and also adaptively estimates the unknown interference in the mathematical model of the quadrotor to improve the control performance. Through theoretical derivation and simulation experiments, this study verified that the proposed algorithm can make the planned trajectory of the quadrotor formation avoid obstacles and make the error between the true trajectory and the planned trajectory converge within a predetermined time under the premise of adaptive estimation of unknown interference in the quadrotor model.

## 1. Introduction

In recent years, remarkable progress has been made in the study of quadrotor formation [1,2,3,4,5,6]. The so-called “quadrotor formation” means that multiple quadrotors complete corresponding tasks according to the expected formation. Among them, how to make the quadrotor formation form the expected formation trajectory within a certain time is an important research topic in the field of multi-agent [4]. In the research field of quadrotor formation, path planning and precise trajectory tracking are the two main research topics. Generally, it is necessary to use trajectory planning to plan the flight path of a quadrotor formation, first to achieve the obstacle avoidance goal, and then use the corresponding control algorithm to track the planned path. The two research fields complement each other and have a certain logical correlation.

For the maintenance of the quadrotor formation, the leader-follower strategy [7] and virtual structure strategy [8] are generally adopted. The leader-follower strategy control is relatively simple, but the follower is dependent on the leader’s trajectory, is unable to make independent decisions, and requires a large amount of communication data. For the virtual structure strategy, the whole formation is regarded as a virtual rigid body structure. Compared with the leader-follower strategy, the follower has less dependence on the leader while using the virtual structure strategy, but the communication data volume is still larger.

In order to further improve the application range and safety of quadrotor formation, obstacle avoidance is an essential and necessary research topic. In order to realize the optimal obstacle avoidance of a quadrotor formation, it is required to plan the path of the formation so that it can avoid obstacles. Currently, the most crucial obstacle avoidance methods of quadrotor formation are obstacle avoidance by using deep reinforcement learning training [9] and obstacle avoidance by using the artificial potential field method [10,11]. For the obstacle avoidance algorithm based on reinforcement learning, the primary method is to train the relevant parameters of the neural network in deep reinforcement learning offline so that each quadrotor in the quadrotor formation can autonomously avoid obstacles. The main shortcoming of this method is that it needs to use as many scenes as possible for a long time of training to achieve the expected obstacle avoidance effect. The obstacle avoidance effect relies on reinforcement learning state spaces, reward functions, and action constraints. The artificial potential field method’s main idea is to take the obstacle as a high potential energy point and, under the repulsive force of the high potential energy point, to plan a path that can ideally bypass the obstacle. Relative to the obstacle avoidance algorithm based on reinforcement learning, there is no need for early training. However, at present, the artificial potential field method for obstacle avoidance in quadrotor formation fields has not solved the local optimal problem.

In order to make the quadrotor track the planned formation trajectory stably and quickly, the stable position and attitude control of the quadrotor are also significant research topics. Currently, many control algorithms have been proposed in the field of quadrotor control, which are mainly divided into linear control and nonlinear control algorithms. Among them, linear control algorithms mainly include the output feedback algorithm [12] and the PID control algorithm. However, the linear control algorithm has certain limitations for multi-freedom, nonlinear-controlled objects such as the quadrotor. Therefore, to improve the controller’s application range, researchers began to study the nonlinear control algorithm in the field of quadrotor control. Various classical nonlinear control algorithms have appeared, such as backstepping control [13], sliding mode control algorithm [13], etc. However, the above control algorithms can only ensure stable convergence of the quadrotor formation control system. Still, they cannot guarantee that the state of the quadrotor system can converge within a finite time. Therefore, a finite-time control algorithm is presented. For example, in the literature [14], the finite-time control algorithm is used to control the quadrotor formation and carry out adaptive estimation of the uncertainty of the model. Compared with the finite-time algorithm, the fixed-time control algorithm can limit the stable time of the quadrotor system state to a known and fixed time range. The literature [5] uses an adaptive fixed-time algorithm based on an RBF neural network to control the quadrotor. The algorithm can make the quadrotor formation converge to the given input in a fixed time, but the specific convergence time cannot be expressed explicitly by the parameters. In recent years, the predefined-time control algorithm has been proposed with the development of nonlinear control fields. The algorithm can make the control system converge within a predetermined time range. Compared with the fixed-time control algorithm, the convergence time of the algorithm can be expressed explicitly as a particular parameter. However, so far, finite-time control algorithms [15,16] and fixed-time control algorithms [5] are primarily used for position control and attitude control of quadrotors. In this paper, the predefined-time control algorithm is extended to control the position loop and attitude loop of the quadrotor so that the quadrotor can be stable within the expected time.

For the nonlinear control of the position and attitude of the quadrotor, the actual control effect depends on the accuracy of the quadrotor mathematical model. However, it is difficult to establish an accurate mathematical model for the quadrotor, and unknown noise interference exists in the environment, so the influence of imprecise modeling in the quadrotor model and unknown interference in the environment on the quadrotor control needs to be adaptive compensation. So far, many algorithms can compensate for environmental interference and imprecise modeling of models, such as the adaptive finite-time control algorithm [15,16] and the adaptive fixed-time control algorithm [5]. However, for the control algorithm of predefined time control, whose convergence time can be explicitly determined by a certain parameter, there is no method to combine it with adaptive algorithms such as the RBF neural network.

Based on the above discussion, this paper studies an artificial potential field method with virtual force for trajectory planning of quadrotor formation and an adaptive predefined-time sliding mode control algorithm based on RBF neural networks to realize the position and attitude control of the quadrotor. The main contributions of this paper are as follows:(1)An artificial potential field method with virtual force is proposed to solve the local optimal problem encountered by using the artificial potential field method in the field of quadrotor formation, and the planned trajectory is input to the position controller of the quadrotor.(2)A predefined-time sliding mode control algorithm for controlling the position and attitude of the quadrotor is studied. Compared with the fixed-time sliding mode algorithm [14], the convergence time of this control algorithm can be expressed explicitly by a certain parameter.(3)On the basis of contribution (2), an adaptive predefined-time sliding mode control algorithm based on RBF neural networks is proposed so that the predefined-time sliding mode control algorithm can be applied to the occasions where there is interference in the environment or inaccurate modeling of the quadrotor model.

## 2. Necessary Preliminaries and Problem Formulation

### 2.1. Necessary Preliminaries

**Lemma** **1**([17])**.** For system $\dot{x}=f\left(t,x,d\right)$, if there exists a radially unbounded Lyapunov function:
(1)$$\dot{V}\left(x\right)\leq -\frac{\pi }{\eta T_{c}\sqrt{\alpha \beta }}\left(\alpha V^{1-\frac{\eta }{2}}+\beta V^{1+\frac{\eta }{2}}\right)+\varepsilon$$

For any state $x\left(t,y_{0}\right)$ of the above system, where $T_{c}>0$ is the predefined time, α,β>0,η∈(0,1) and 0≤ε≤∞ is the system parameter, then the motion path x of the system state is stable in the predefined time, and the residual set of the solution of the system can be written by:(2)$$\{\underset{t\to T_{pc}}{\lim}x|V\left(x\right)\leq \min\{{\left(\frac{\varepsilon \eta T_{c}\sqrt{\alpha \beta }}{\pi \alpha \left(1-\mu \right)}\right)}^{\frac{2}{2-\eta }},{\left(\frac{\varepsilon \eta T_{c}\sqrt{\alpha \beta }}{\pi \alpha \left(1-\mu \right)}\right)}^{\frac{2}{2+\eta }}\}\}$$
where 0<μ<1 and the predefined time $T_{pc}=T_{c}/\sqrt{\mu }$.

**Lemma** **2**([18])**.** When y≥x, and ϖ>1, then:
(3)$$x{\left(y-x\right)}^{\bar{\omega }}\leq \frac{\varpi }{1+\varpi }\left(y^{1+\varpi }-x^{1+\varpi }\right)$$

**Lemma** **3**([19])**.** For all positive $x_{i}\left(i=1,2,\ldots ,n\right)$ and γ>0, we get:
(4)$$\left\{ \begin{array}{l} \sum_{i=1}^{n}x_{i}^{\gamma }\geq {\left(\sum_{i=1}^{n}x_{i}\right)}^{\gamma },if\;0<\gamma <1\\ \sum_{i=1}^{n}x_{i}^{\gamma }\geq n^{1-\gamma }{\left(\sum_{i=1}^{n}x_{i}\right)}^{\gamma },if\;\gamma >1 \end{array}\right.$$

**Lemma** **4**([18])**.** For x∈R,y∈R and positive numbers p>0, we get:
(5)$${\left|x\right|}^{p}{\left|y\right|}^{q}\leq \frac{p}{p+q}\delta {\left|x\right|}^{p+q}+\frac{q}{p+q}\delta^{-\frac{p}{q}}{\left|y\right|}^{p+q}$$

### 2.2. Quadrotor Dynamic Model

Before describing the dynamics and kinematics models of the quadrotor, the inertial coordinate system and the body coordinate system are briefly introduced and the relationship between the quadrotor model and the coordinate axis is shown in the Figure 1:

1. Inertial coordinate system $O_{xyz}$: the origin of coordinates O is at a point on the surface of the earth. The axis $O_{x}$ is on the ground plane, pointing east. The axis $O_{z}$ is negative and perpendicular to the ground plane, pointing to the center of the earth. The axis $O_{y}$, the axis $O_{x}$, and the axis $O_{z}$ constitute the right-hand coordinate system.

2. Body coordinate system $o_{i,xbybzb}$: the origin of coordinates o is located at the center of the mass of the quadrotor. The axis $o_{i,xb}$ coincides with the movement direction of the quadrotor. The axis $o_{i,zb}$ is perpendicular to the plane of the quadrotor body. The axis $o_{i,yb}$, the axis $o_{i,xb}$, and the axis $o_{i,zb}$ constitute the right-hand coordinate system. The coordinate system represents the body coordinate system of the *i*th quadrotor.

The transformation matrix of the quadrotor from the inertial coordinate system $O_{xyz}$ to the volume coordinate system $o_{i,xbybzb}$ is expressed as:(6)$$R_{i}^{I\to B}=\left[\begin{array}{ccc} \cos\theta \cos\psi & \cos\theta \sin\psi & -\sin\theta \\ \sin\varphi \sin\theta \cos\psi -\cos\varphi \sin\psi & \sin\varphi \sin\theta \sin\psi +\cos\varphi \cos\psi & \sin\phi \cos\theta \\ \cos\varphi \sin\theta \cos\psi +\sin\varphi \sin\psi & \cos\varphi \sin\theta \sin\psi -\sin\varphi \cos\psi & \cos\phi \cos\theta \end{array}\right]$$
where θ,φ,ψ, respectively, represent the roll angle, pitch angle, and yaw angle of the quadrotor.

The mathematical model of a quadrotor is divided into two parts: the position loop and the attitude loop. Based on the inertial coordinate system, the quadrotor position loop model is shown as follows:(7)$$\left[\begin{array}{c} {\ddot{x}}_{i}\\ {\ddot{y}}_{i}\\ {\ddot{z}}_{i} \end{array}\right]=-\left[\begin{array}{c} 0\\ 0\\ g \end{array}\right]+\frac{1}{m_{i}}R_{i}^{B\to I}\left[\begin{array}{l} 0\\ 0\\ T_{i} \end{array}\right]+\left[\begin{array}{l} -\frac{k_{i,x}}{m_{i}}{\dot{x}}_{i}\\ -\frac{k_{i,y}}{m_{i}}{\dot{y}}_{i}\\ -\frac{k_{i,z}}{m_{i}}{\dot{z}}_{i} \end{array}\right]-\left[\begin{array}{l} 0\\ 0\\ f_{i,z} \end{array}\right]$$where, i represents the ith quadrotor; ${\ddot{x}}_{i},{\ddot{y}}_{i},{\ddot{z}}_{i}$ represent the acceleration of the ith quadrotor in the x,y,z directions; g is the gravitational acceleration; $m_{i}$ represents the mass of the ith quadrotor; $R_{i}^{B\to I}$ is the inverse matrix of $R_{i}^{I\to B}$; $T_{i}$ is the lift force of the ith quadrotor; $k_{i,x},k_{i,y},k_{i,z}$ are the drag coefficients in the x,y,z directions, respectively, in the inertial coordinate system; ${\dot{x}}_{i},{\dot{y}}_{i},{\dot{z}}_{i}$ represent the speed of the ith quadrotor in x,y,z directions; $f_{i,z}$ is the unknown interference in the z axis direction.

For the convenience of expression, Equation (7) is converted into vector form in this paper and replaced as follows:

$P_{i}={\left[x_{i},y_{i},z_{i}\right]}^{T}$; $G={\left[0,0,g\right]}^{T}$; $F_{i}={\left[0,0,f_{i,z}\right]}^{T}$; $U_{i}=\frac{1}{m_{i}}R^{B\to I}[0,{0,T_{i}]}^{T}$; $K_{i}=$${\left[-\frac{k_{i,x}}{m_{i}},-\frac{k_{i,y}}{m_{i}},-\frac{k_{i,z}}{m_{i}}\right]}^{T}$. Then Formula (7) can be rewritten as:(8)$${\overset{¨}{P}}_{i}=-G+U_{i}+K_{i}{\dot{P}}_{i}+F_{i}$$

The quadrotor attitude model is shown as follows: (9)$$\left[\begin{array}{l} {\overset{¨}{\theta }}_{i}\\ {\overset{¨}{\varphi }}_{i}\\ {\overset{¨}{\psi }}_{i} \end{array}\right]=\left[\begin{array}{l} \tau_{i,\theta }\\ \tau_{i,\varphi }\\ \tau_{i,\psi } \end{array}\right]$$

This paper makes the following substitutions: $A_{i}={\left[\theta_{i},\varphi_{i},\psi_{i}\right]}^{T}$ and $\tau_{i}=\left[\tau_{i,\theta },\tau_{i,\varphi },\tau_{i,\psi }\right]$, then the Formula (9) can be written as Formula (10):(10)$${\overset{¨}{A}}_{i}=\tau_{i}$$
where $A_{i}$ represents the attitude of the *i*th quadrotor (the vector composed of roll angle $\theta_{i}$, pitch angle $\varphi_{i}$ and yaw angle $\psi_{i}$), and $\tau_{i}$ represents the torque generated in the attitude direction of the quadrotor.

The given value of the quadrotor attitude loop is calculated from the control quantity of the position loop, where, $\phi_{i}^{d}$ denotes the expected value of the attitude angle $\phi_{i}$, $\theta_{i}^{d}$ denotes the expected value of the attitude angle $\theta_{i}$, and $\psi_{i}^{d}$ denotes the expected value of the attitude angle $\psi_{i}$. The specific formula is shown as follows:(11)$$\left\{ \begin{array}{l} \phi_{i}^{d}=\arcsin\left(m_{i}u_{i,x}\sin\psi_{i}^{d}-u_{i,y}\cos\psi_{i}^{d}/T_{i}\right)\\ \theta_{i}^{d}=\arctan\left(u_{i,x}\cos\psi_{i}^{d}+u_{i,y}\sin\psi_{i}^{d}/u_{i,z}\right)\\ \psi_{i}^{d}=0 \end{array}\right.$$
where $u_{i,x},u_{i,y},u_{i,z}$, respectively, represent the component force of lift force $T_{i}$ in the direction x,y,z, and the conversion relationship is expressed by the following formula:(12)$$\left\{ \begin{array}{l} u_{i,x}=\frac{T_{i}}{m_{i}}\left(\cos\psi_{i}\sin\theta_{i}\cos\phi_{i}+\sin\psi_{i}\sin\phi_{i}\right)\\ u_{i,y}=\frac{T_{i}}{m_{i}}\left(\sin\psi_{i}\sin\theta_{i}\cos\phi_{i}-\cos\psi_{i}\cos\phi_{i}\right)\\ u_{i,z}=\frac{T_{i}}{m_{i}}\cos\theta_{i}\cos\phi_{i} \end{array}\right.$$

### 2.3. RBF Neural Network Estimation

RBFNN is a very effective tool for the accurate estimation of unknown interference. In 1988, Broomhead and Lowe introduced RBF into neural network design based on the local response of biological neurons. In 1989, Jackson demonstrated the uniform approximation performance of the RBF neural network for nonlinear continuous functions. RBFNN is a very effective tool for the accurate estimation of unknown interference. The literature [20] compares the performances of the RBF neural network and the BP neural network in approximating nonlinear functions, and the research shows that the generalization ability of the RBF neural network is superior to the BP neural network in all aspects. In this paper, RBFNN will be used for the adaptive estimation of unknown interference $f_{i,z}$ from the z axis of the *i*th quadrotor. Its network structure is shown in Figure 2:

The first layer (input layer): $I_{i}={\left[i_{1},i_{2},\ldots ,i_{m}\right]}^{T}$ represents the input of the neural network, and m represents the input dimension.

The second layer (hidden layer): the output is $h\left(I_{i}\right)={\left[h_{1},\ldots ,h_{n}\right]}^{T}$, and the Gaussian base is used as the membership function of the input layer, namely: $h_{j}\left(I_{i}\right)=\exp\left(-\frac{{\left\|I_{i}-C_{i,j}\right\|}^{2}}{2b_{i}{}^{2}}\right)$, where, $b_{i}$ is the width of the neural network (the mean of the Gaussian basis function), and $C_{i,j}={\left[c_{i,j,1},c_{i,j,2},\ldots ,c_{i,j,n}\right]}^{T}$ is the center of the neural network (the variance of the Gaussian basis function).

The third layer (output layer): the output of the neural network is: $y=W^{T}h\left(I_{i}\right)=w_{1}h_{1}+\ldots +w_{n}h_{n}$, where, W is the weight of RBF neural network.

The unknown interference can be expressed by Equation (13):(13)$$f_{i,z}=W_{i}^{*T}h\left(I_{i}\right)-\varepsilon_{i,z}$$

Moreover, the estimated value $f_{i,z}$ of unknown interference can be expressed by Equation (14):(14)$${\hat{f}}_{i}=\hat{W_{i}^{T}}h\left(I_{i}\right)$$
where $\varepsilon_{i,z}\left(\left|\varepsilon_{i,z}\right|\leq \varepsilon_{N},\varepsilon_{N}\geq 0\right)$ is the network approximation error, $W_{i}^{*}={\left[w_{i,1}^{*},w_{i,2}^{*},\ldots ,w_{i,n}^{*}\right]}^{T}$ is the real weight of the neural network, ${\hat{W}}_{i}={\left[{\hat{w}}_{i,1},{\hat{w}}_{i,2},\ldots ,{\hat{w}}_{i,n}\right]}^{T}$ represents the estimated value of the real weight, and $h\left(I_{i}\right)={\left[h_{1}\left(I_{i}\right),h_{2}\left(I_{i}\right),\ldots ,h_{n}\left(I_{i}\right)\right]}^{T}$ is the RBF vector of the hidden layer.

## 3. Path Planning and Controller Design

Design objectives: Firstly, this paper uses the artificial potential field method to convert the ideal trajectory of each quadrotor in the formation into the planned trajectory, so as to achieve the target of collision avoidance between the formation and obstacles as well as between each quadrotor in the formation, and then inputs the planned trajectory to the controller of each quadrotor in the formation. The controller is combined with an RBF neural network to estimate the unknown interference of each quadrotor in the z axis direction. Finally, the predefined-time sliding mode control algorithm is used to make each quadrotor stably track the planned trajectory within the predefined time. The design flow chart is shown in Figure 3:

### 3.1. The Path Planning of the Formation

The overall design idea of formation trajectory planning in this paper is as follows: according to the preset ideal trajectory of the ith quadrotor $\xi_{i,1}^{r}={\left[x_{i}^{r},y_{i}^{r},z_{i}^{r}\right]}^{T}$, combined with the planned trajectory of other quadrotors $\xi_{j,1}^{d}={\left[x_{j}^{d},y_{j}^{d},z_{j}^{d}\right]}^{T}$, and the position of the ith obstacle $O_{k}\left(k=0,1,2,\ldots ,m\right)$, this paper uses the artificial potential field method to calculate the planned trajectory of the ith quadrotor $\xi_{i.1}^{d}={\left[x_{i}^{d},y_{i}^{d},z_{i}^{d}\right]}^{T}$.

The actual planned path is shown in Equation (15), and the control quantity is shown in Equation (16) [21]:(15)$$\left\{ \begin{array}{l} \dot{\xi_{i,1}^{d}}=\xi_{i,2}^{d}\\ \dot{\xi_{i,2}^{d}}=\Gamma_{i} \end{array}\right.$$
(16)$$\Gamma_{i}=c_{i,1}\left(\xi_{i,1}^{r}-\xi_{i,1}^{d}\right)+c_{i,2}\left(\xi_{i,2}^{r}-\xi_{i,2}^{d}\right)+\varsigma_{i}+\varsigma_{i,o}+\iota_{i}+\iota_{i,o}$$
where $\varsigma_{i}$ denotes that the *i*th quadrotor is subjected to the vector sum of the artificial potential field repulsive force of all other quadrotors, $\varsigma_{i,o}$ denotes that the *i*th quadrotor is subjected to the vector sum of the artificial potential field repulsive force of all other obstacles, and $\iota_{i},\iota_{i,o}$ is the virtual force preventing the artificial potential field from falling into the local optimal.

In order to prevent the artificial potential field method between quadrotors and collisions from falling into the local optimal solution, when $\left|∠\left(\xi_{i,2}^{d},\varsigma_{i}\right)\right|<\alpha$, this paper sets: $\left\|\iota_{i}\right\|=\left\|\varsigma_{i}\right\|,\iota_{i}⊥\varsigma_{i}$, and set $c_{1}=c_{2}=0$ to temporarily avoid the approximation of the planned trajectory to the ideal trajectory, so as to prevent the quadrotor from falling into the local optimal solution by adding virtual force. When the local optimum is removed ($\left|∠\left(\xi_{i,2}^{d},\varsigma_{i}\right)\right|\geq \alpha$), the value of $c_{1},c_{2}$ is restored, and the virtual force $\iota_{i}$ is removed.

In order to prevent the artificial potential field method between formation and falling into the local optimal solution, when $\left|∠\left(\xi_{i,2}^{d},\varsigma_{i,o}\right)\right|<\beta$, this paper sets: $\left\|\iota_{i,o}\right\|=\left\|\varsigma_{i,o}\right\|,\iota_{i,o}⊥\varsigma_{i,o}$, and sets $c_{1}=c_{2}=0$ to temporarily avoid the approximation of the planned trajectory to the ideal trajectory, so as to prevent the quadrotor from falling into the local optimal solution by adding virtual force. When the local optimum is removed ($\left|∠\left(\xi_{i,2}^{d},\varsigma_{i,o}\right)\right|\geq \beta$), the value of $c_{1},c_{2}$ is restored, and the virtual force $\iota_{i,o}$ is removed.

The purpose of the angles that limit the values of $\left|∠\left(\xi_{i,2}^{d},\varsigma_{i}\right)\right|$ and $\left|∠\left(\xi_{i,2}^{d},\varsigma_{i,o}\right)\right|$ is to determine whether it will fall into the local optimum because of the repulsive force generated by the artificial potential field. This angle can be selected arbitrarily in theory, but when the angle is too small, the calculated obstacle avoidance path may not be smooth and continuous. If the selected angle is too large, unnecessary obstacle avoidance actions will be executed when the local optimization does not appear. Finally, through simulation experiments, we set the values of α and β to $10^{\circ }$.

The artificial potential field function is defined as follows:(17)$$\varphi \left(X\right)=\left\{ \begin{array}{l} \frac{1}{2}k_{\varsigma }\left(\frac{1}{X}-\frac{1}{R_{\varsigma }}\right),0<X\leq R_{\varsigma }\\ 0,X>R_{\varsigma } \end{array}\right.$$
where $R_{\varsigma }$ is the intensity radius of the artificial potential field, φ(X) is the artificial potential function, and $k_{\varsigma }$ is the intensity coefficient of the artificial potential field.

The artificial potential field between the quadrotors in the formation is described below. Assuming that the *i*th quadrotor is located in the artificial potential field generated by other nearby quadrotors, the repulsive force of the potential field generated by other quadrotors and the sum of the artificial potential field strength of the *i*th quadrotor are shown in Formula (18) and Formula (19), respectively:(18)$$\varsigma_{i}=-\nabla_{\xi_{i,1}^{d}}V\left(\bar{\xi_{i}}\right)$$
(19)$$V\left(\bar{\xi_{i}}\right)=\sum_{j\neq i,j=1}^{n}\varphi \left(\left\|\xi_{j,1}^{d}-\xi_{i,1}^{d}\right\|\right),i=0,1,2,\ldots ,n$$
where $\varsigma_{i}$ represents the sum of the repulsive force vector of the artificial potential field of the *i*th quadrotor subjected to all other quadrotors, $V\left({\bar{\xi }}_{i}\right)$ represents the sum of the artificial potential field intensity between the *i*th quadrotor and all other quadrotors, and $\nabla_{\xi_{i,1}^{d}}$ is the negative gradient term.

The artificial potential field between the formation and the obstacle is described below. The obstacle avoidance area $R_{k}$ is defined near the kth obstacle and $R_{k}$ indicates the obstacle avoidance radius. When the *i*th quadrotor approaches the obstacle, the intersection point of the line between the obstacle k and the quadrotor i and the sphere formed by the obstacle avoidance radius $R_{k}$ is defined as:(20)$$\xi_{i,k,1}^{d}=O_{k}+R_{k}\frac{\xi_{i,1}^{d}-O_{k}}{\left\|\xi_{i,1}^{d}-O_{k}\right\|}$$
where $O_{k}={\left(x_{ok},y_{ok},z_{ok}\right)}^{T}$ is the coordinate of the kth obstacle. The collision avoidance control items between the *i*th quadrotor and obstacles are shown in Formula (21):(21)$$\varsigma_{i,o}=-\nabla_{\xi_{i,1}^{d}}V\left({\bar{\xi }}_{i,o}\right)$$
(22)$$V\left({\bar{\xi }}_{i,o}\right)=\sum_{k=1}^{n}\varphi \left(\left\|\xi_{i,k,1}^{d}-\xi_{i,1}^{d}\right\|\right),i=0,1,2,\ldots ,n$$
where $\varsigma_{i,o}$ denotes the sum of the repulsive force vector of the artificial potential field of the *i*th quadrotor subjected to all other obstacles and $V\left({\bar{\xi }}_{i,o}\right)$ denotes the sum of the artificial potential field strength between the *i*th quadrotor and all other obstacles.

Through the above formulas, each planned trajectory of the quadrotor within the formation can avoid other quadrotors and obstacles. The control items $\varsigma_{i}$ and $\varsigma_{i,k}$ in Equation (16) are eliminated in the final stable non-obstacle avoidance region. Finally, a linear system is obtained as follows:(23)$$\overset{¨}{\xi_{i,1}^{d}}+c_{2}\dot{\xi_{i,1}^{d}}+c_{1}\xi_{i,1}^{d}=c_{2}\dot{\xi_{i,1}^{r}}+c_{1}\xi_{i,1}^{r}$$

If the two parameters $c_{1}$ and $c_{2}$ are correctly selected $\left(c_{1}<c_{2}\right)$, the system will be stable ($\xi_{i,1}^{d}-\xi_{i,1}^{r}$ will approach 0), that is, in the non-obstacle avoidance region, the planned trajectory $\xi_{j,1}^{d}$ can stably track the ideal trajectory $\xi_{i,1}^{r}$.

### 3.2. Controller Design

The control of a quadrotor is mainly divided into two parts: the position control and the attitude control. Firstly, this paper controls the position loop by an adaptive predefined-time sliding mode controller based on an RBF neural network according to the planned path $\xi_{i,1}^{d}$ of the obstacle avoidance calculated in the previous step. Then, the control output of the position loop is input into the attitude loop, and the attitude of the quadrotor is controlled through the predefined-time sliding mode control of the attitude loop so as to finally track the specified formation according to the planned path $\xi_{i,1}^{d}$.

#### 3.2.1. Predefined-Time Sliding Mode Controller Design of the Position Loop

In this section, through theoretical analysis, this paper proves that the error between the real position and the expected position of the quadrotor can be stabilized within a predefined time. The proof process is described as follows:

In order to prove that the position loop error can converge stably, the Lyapunov function is defined:(24)$$V_{1,i}=\frac{1}{2}E_{i,P}^{T}E_{i,P}$$
where $E_{i,p}=P_{i}-\xi_{1,i}^{d}$ represents the position error of the quadrotor. The sliding mode switch function is defined as:(25)$$S_{i,P}=\dot{E_{i,P}^{}}+\Psi_{i}$$

Among them, $\Psi_{i}=\frac{\pi }{2\eta_{1,i}T_{c1,i}\sqrt{\alpha_{1,i}\beta_{1,i}}}(\alpha_{1,i}V_{1,i}^{-\eta_{1,i}/2}+\beta_{1,i}V_{1,i}^{\eta_{1,i}/2})E_{i,p}$, $\alpha_{1,i},\beta_{1,i}$ and $0<\eta_{1,i}<1$ are three positive numbers. $T_{c1,i}$ is the predefined time taken for the sliding mode surface to converge to 0.

For Equation (25), when the sliding mode surface $S_{p,i}$ converges to 0, Equation (25) is reduced to Equation (26):(26)$${\dot{E}}_{i,p}=-\Psi_{i}$$

Thus, by taking the derivative of Equation (24) and substituting Equation (26) into the derivative of Equation (24), we can get:(27)$$\begin{array}{ll} {\dot{V}}_{1,i} & =-E_{i,P}^{T}\Psi_{i}\\ & \begin{array}{c} =-\frac{\pi }{2\eta_{1,i}T_{c1,i}\sqrt{\alpha_{1,i}\beta_{1,i}}}\left(\alpha_{1,i}V_{1,i}^{1-\eta_{1,i}/2}+\beta_{1,i}V_{1,i}^{1+\eta_{1,i}/2}\right) \end{array} \end{array}$$

According to Lemma 1, $E_{i}$ will converge to 0 in a predefined time $T_{c1,i}$. Next, this article will prove that the sliding mode surface $S_{i,P}$ will also converge to 0 in a predefined time $T_{c2,i}$.

For the sliding mode switch function $S_{i,P}$, in order to prove that the sliding mode switch function can also converge stably to 0, this paper defines the Lyapunov function:(28)$$V_{2,i}=\frac{1}{2}S_{i,P}^{T}S_{i,P}$$

By substituting Equation (25) into the derivative of Equation (28), we can get:(29)$$\begin{array}{ll} {\dot{V}}_{2,i} & =S_{i,P}^{T}{\dot{S}}_{i,P}\\ & \begin{array}{c} =S_{i,P}^{T}\left(K_{i}{\dot{P}}_{i}+U_{i}+F_{i}-G+{\dot{\Psi }}_{i}-{\ddot{P}}_{i}^{d}\right) \end{array} \end{array}$$

Let the control quantity $U_{i}$ be:(30)$$\begin{array}{l} U_{i}=-\frac{\pi }{2\eta_{2,i}T_{c2,i}\sqrt{\alpha_{2,i}\beta_{2,i}}}(\alpha_{2,i}V_{2,i}^{-\eta_{2,i}/2}+2^{\eta_{2,i}/2}\beta_{2,i}V_{2,i}^{\eta_{2,i}/2})S_{p,i}\\ \begin{array}{cc} & +\overset{¨}{P_{i}^{d}}-K_{i}{\dot{P}}_{i}+G-{\dot{\Psi }}_{i}-{\hat{F}}_{i}+\varepsilon_{N}Isign\left(S_{i,P}\right) \end{array} \end{array}$$
where $\alpha_{2,i},\beta_{2,i},\varepsilon_{N}$, and $0<\eta_{2,i}<1$ are four positive numbers, and I is the identity matrix. Substituting (30) into (29), we can get:(31)$${\dot{V}}_{2,i}=-\frac{\pi }{\eta_{2,i}T_{c2,i}\sqrt{\alpha_{2,i}\beta_{2,i}}}(\alpha_{2,i}V_{2,i}^{1-\eta_{2,i}/2}+2^{\eta_{2,i}/2}\beta_{2,i}V_{2,i}^{1+\eta_{2,i}/2})+S_{i,P}^{T}\left(F_{i}-{\hat{F}}_{i}\right)+\left|S_{i,P}^{T}S_{i,P}^{}\right|\varepsilon_{N}$$
where $F_{i}={\left[0,0,f_{i,z}\right]}^{T}$ represents the unknown interference in the position model of the quadrotor, and ${\hat{F}}_{i}={\left[0,0,{\hat{f}}_{i,z}\right]}^{T}$ represents the compensation of the quadrotor model controller to the unknown interference. Therefore, by substituting (13) and (14) into (31), we can get:(32)$${\dot{V}}_{2,i}=-\frac{\pi }{\eta_{2,i}T_{c2,i}\sqrt{\alpha_{2,i}\beta_{2,i}}}(\alpha_{2,i}V_{2,i}^{1-\eta_{2,i}/2}+2^{\eta_{2,i}/2}\beta_{2,i}V_{2,i}^{1+\eta_{2,i}/2})+S_{i,P}^{T}H_{i}^{T}\tilde{W_{i}}+\kappa_{i}$$

For convenience, let $\kappa_{i}=\left|S_{i,P}^{T}S_{i.P}^{}\right|\varepsilon_{N}-S_{i,P}^{T}\varepsilon_{i}$, and $\varepsilon_{i}={\left[0,0,\varepsilon_{i,z}\right]}^{T}$ is the error between the estimated value and the real value of the neural network in the quadrotor position model. ${\tilde{W}}_{i}=W_{i}^{*}-{\hat{W}}_{i}$ represents the difference between the estimated weight value and the real weight value of the neural network. The hidden layer output is: $H_{i}=\left[0,0,h\left(I_{i}\right)\right]$. Since the difference ${\tilde{W}}_{i}$ of adaptive weight value is added, in order to prove that ${\tilde{W}}_{i}$ will converge to 0 in the predefined time, Lyapunov function $V_{3,i}$ is defined again:(33)$$V_{3,i}=V_{2,i}+\frac{1}{2}{\tilde{W}}_{i}^{T}{\tilde{W}}_{i}$$

By differentiating Equation (33), we can get:(34)$${\dot{V}}_{3,i}=-\frac{\pi }{\eta_{2,i}T_{c2,i}\sqrt{\alpha_{2,i}\beta_{2,i}}}(\alpha_{2,i}V_{2,i}^{1-\eta_{2,i}/2}+2^{\eta_{2,i}/2}\beta_{2,i}V_{2,i}^{1+\eta_{2,i}/2})+S_{i,P}^{T}H_{i}^{T}{\tilde{W}}_{i}+\kappa_{i}-\dot{{\hat{W}}_{i}^{T}}{\tilde{W}}_{i}$$

Let the adaptive rate ${\dot{\hat{W}}}_{i}$ be:(35)$${\dot{\hat{W}}}_{i}=H_{i}S_{i,P}-{\left(\frac{1}{2}\right)}^{1+\eta_{2,i}/2}k_{2,i}2^{\eta_{2,i}/2}\beta_{2,i}C_{i}n^{\eta_{2,i}/2}{\left({\hat{W}}_{i}^{}\right)}^{1+\eta_{2,i}}-\alpha_{2,i}\upsilon_{i}{\hat{W}}_{i}$$

Among them, $k_{2,i}=\frac{\pi }{\eta_{2,i}T_{c2,i}\sqrt{\alpha_{2,i}\beta_{2,i}}}$, $C_{i}=\frac{2+\eta_{2,i}}{1+\eta_{2,i}}$, $\upsilon_{i}=$$k_{2,i}^{2/\left(2-\eta_{2,i}\right)}/2$, and n represent the dimension of the weight ${\hat{W}}_{i}$ of the neural network.

By substituting Equation (35) into Equation (34), we get:(36)$$\begin{array}{ll} {\dot{V}}_{3,i} & =-k_{i}\alpha_{2,i}V_{2,i}^{1-\eta_{2}/2}-k_{i}2^{\eta_{2,i}/2}\beta_{2,i}V_{2,i}^{1+\eta_{2}/2}+\alpha_{2,i}\upsilon_{i}{\hat{W}}_{i}{}^{T}{\tilde{W}}_{i}\\ & \begin{array}{c} +{\left(\frac{1}{2}\right)}^{1+\eta_{2,i}/2}k_{i}2^{\eta_{2,i}/2}\beta_{2,i}C_{i}n^{\eta_{2,i}/2}{\left({\hat{W}}_{i}^{T}\right)}^{1+\eta_{2}}{\tilde{W}}_{i}+\kappa_{i} \end{array} \end{array}$$

According to Young’s inequality, we can get:(37)$${\hat{W}}_{i}^{T}{\tilde{W}}_{i}={\tilde{W}}_{i}^{T}\left(W_{i}^{*}-\tilde{W}\right)\leq -{\tilde{W}}_{i}^{T}{\tilde{W}}_{i}+W_{i}^{*}{}^{T}W_{i}^{*}$$

By substituting Equation (37) into Equation (36), we can get:(38)$$\begin{array}{ll} {\dot{V}}_{3,i} & =-k_{i}\alpha_{2,i}V_{2,i}^{1-\eta_{2}/2}-k_{i}2^{\eta_{2,i}/2}\beta_{2,i}V_{2,i}^{1+\eta_{2}/2}-\alpha_{2,i}\upsilon_{i}{\tilde{W}}_{i}^{T}{\tilde{W}}_{i}\\ & \begin{array}{c} +{\left(\frac{1}{2}\right)}^{1+\eta_{2,i}/2}k_{i}2^{\eta_{2,i}/2}\beta_{2,i}C_{i}n^{\eta_{2,i}/2}{\left({\hat{W}}_{i}^{T}\right)}^{1+\eta_{2}}{\tilde{W}}_{i}+\alpha_{2,i}\upsilon_{i}W_{i}^{*T}W_{i}^{*}+\kappa_{i} \end{array} \end{array}$$

By using lemma 4 and setting $x=1,y=\alpha_{2,i}\upsilon_{i}{\tilde{W}}_{i}^{T}\tilde{W},p=\eta_{2,i}/2,q=1-\eta_{2,i}/2$ and δ= $\exp\left(\left(2-\eta_{2,i}\right)/\eta_{2,i}\right)\ln\left(2-\eta_{2,i}\right)/2$, we get:(39)$${\left(\upsilon_{i}{\tilde{W}}_{i}^{T}{\tilde{W}}_{i}\right)}^{1-\eta_{2,i}/2}\leq \upsilon_{i}{\tilde{W}}_{i}^{T}{\tilde{W}}_{i}+ο\left(\eta_{2,i}\right)$$

By substituting Equation (39) into Equation (38), we can get:(40)$$\begin{array}{l} {\dot{V}}_{3,i}=-k_{i}\alpha_{2,i}V_{2,i}^{1-\eta_{2,i}/2}-k_{i}2^{\eta_{2,i}/2}\beta_{2,i}V_{2,i}^{1+\eta_{2,i}/2}-k_{i}\alpha_{2,i}{\left(\frac{1}{2}{\tilde{W}}_{i}^{T}{\tilde{W}}_{i}\right)}^{1-\eta_{2,i}/2}+{\left(\frac{1}{2}\right)}^{1+\eta_{2,i}/2}k_{i}2^{\eta_{2,i}/2}\\ \begin{array}{cc} & \beta_{2,i}C_{i}n^{\eta_{2,i}/2}{\left({\hat{W}}_{i}^{T}\right)}^{1+\eta_{2,i}}{\tilde{W}}_{i}+\alpha_{2,i}\upsilon_{i}W_{i}^{*T}W_{i}^{*}+ο\left(\eta_{2,i}\right)+\kappa_{i} \end{array} \end{array}$$

By proof of Equation (42) and Equation (43), the inequality (41) can be obtained:(41)$$\begin{array}{ll} {\left({\hat{W}}_{i}^{T}\right)}^{1+\eta_{2,i}}{\tilde{W}}_{i} & ={\left[{\left(W_{i}^{*}-{\tilde{W}}_{i}\right)}^{T}\right]}^{1+\eta_{2,i}}{\tilde{W}}_{i}\\ & \leq \frac{1+\eta_{2,i}}{2+\eta_{2,i}}(\sum_{i=1}^{n}{\left(w_{i}^{*}\right)}^{1+\eta_{2,i}/2}-\sum_{i=1}^{n}{\left({\tilde{w}}_{i}{}^{2}\right)}^{1+\eta_{2,i}/2}) \end{array}$$

Next, this paper will prove the inequality in Equation (41), and the proof process is described as follows:

First, the vector form of the following equation is decomposed into the scalar form:(42)$${\left(\frac{1}{2}\right)}^{1+\eta_{2,i}/2}{\left({\hat{W}}_{i}^{T}\right)}^{1+\eta_{2,i}}{\tilde{W}}_{i}={\left(\frac{1}{2}\right)}^{1+\eta_{2,i}/2}\sum_{i=1}^{n}{\tilde{w}}_{i}{\left(w_{i}^{*}-{\tilde{w}}_{i}\right)}^{1+\eta_{2,i}}$$

Finally, from Lemma 2, Equation (42) can be simplified to:(43)$$\begin{array}{ll} {\left(\frac{1}{2}\right)}^{1+\eta_{2,i}/2}\sum_{i=1}^{n}{\tilde{w}}_{i}{\left(w_{i}^{*}-{\tilde{w}}_{i}\right)}^{1+\eta_{2,i}} & \leq {\left(\frac{1}{2}\right)}^{1+\eta_{2,i}/2}\frac{1+\eta_{2,i}}{2+\eta_{2,i}}\left(\sum_{i=1}^{n}w_{i}^{*}{}^{2+\eta_{2,i}}-\sum_{i=1}^{n}{\tilde{w}}_{i}^{2+\eta_{2,i}}\right)\\ & \begin{array}{c} =\frac{1+\eta_{2,i}}{2+\eta_{2,i}}\left(\sum_{i=1}^{n}{\left(\frac{w_{i}^{*}{}^{2}}{2}\right)}^{1+\eta_{2,i}/2}-\sum_{i=1}^{n}{\left(\frac{{\tilde{w}}_{i}^{2}}{2}\right)}^{1+\eta_{2,i}/2}\right) \end{array} \end{array}$$

Therefore, this paper proves that the inequality in Equation (41) is valid.

By substituting inequality (41) into Equation (40), we can get:(44)$$\begin{array}{l} {\dot{V}}_{3,i}\leq -k_{i}\alpha_{2,i}V_{2,i}^{1-\eta_{2,i}/2}-k_{i}2^{\eta_{2,i}/2}\beta_{2,i}V_{2,i}^{1+\eta_{2,i}/2}{}^{1-\eta_{2,i}/2}-k_{i}\alpha_{2,i}\left(\frac{1}{2}{\tilde{W}}_{i}^{T}{\tilde{W}}_{i}\right)-k_{i}2^{\eta_{2,i}/2}\beta_{2,i}n^{\eta_{2,i}/2}\sum_{i=1}^{n}{\left(\frac{{\tilde{w}}_{i}^{2}}{2}\right)}^{1+\eta_{2,i}/2}\\ \begin{array}{cc} & \begin{array}{cc} & +k_{i}2^{\eta_{2,i}/2}\beta_{2,i}n^{\eta_{2,i}/2}\sum_{i=1}^{n}{\left(\frac{w_{i}^{*}{}^{2}}{2}\right)}^{1+\eta_{2,i}/2}+k_{i}\alpha_{2,i}(\frac{1}{2}{\tilde{W}}_{i}^{T}{\tilde{W}}_{i})+ο\left(\eta_{2,i}\right)+\kappa_{i} \end{array} \end{array} \end{array}$$

Let $D_{i}=ο\left(\eta_{2,i}\right)+\kappa_{i}+k_{i}\alpha_{2,i}{\left(\frac{1}{2}{\tilde{W}}_{i}^{T}{\tilde{W}}_{i}\right)}^{1-\eta_{2,i}/2}+k_{i}2^{\eta_{2,i}/2}\beta_{2,i}\sum_{i=1}^{n}{\left(\frac{w_{i}^{*}{}^{2}}{2}\right)}^{1+\eta_{2,i}/2}>0$, then by Lemma 3, Equation (44) can be rewritten as:(45)$$\begin{array}{l} {\dot{V}}_{3,i}\leq -k_{i}\alpha_{2,i}V_{2,i}^{1-\eta_{2,i}/2}-k_{i}2^{\eta_{2,i}/2}\beta_{2,i}V_{2,i}^{1+\eta_{2,i}/2}-k_{i}\alpha_{2,i}{\left(\frac{1}{2}{\tilde{W}}_{i}^{T}{\tilde{W}}_{i}\right)}^{1-\eta_{2,i}/2}\\ \begin{array}{cc} & -k_{i}2^{\eta_{2,i}/2}\beta_{2,i}{\left(\frac{1}{2}{\tilde{W}}_{i}^{T}{\tilde{W}}_{i}\right)}^{1+\eta_{2,i}/2}+D_{i} \end{array}\\ \begin{array}{cc} & = \end{array}-k_{i}\left(V_{3,i}^{1-\eta_{2,i}/2}+V_{3,i}^{1+\eta_{2,i}/2}\right)+D_{i} \end{array}$$

According to Lemma 1, both $S_{i,P}$ and ${\tilde{W}}_{i}$ will approach 0 in the predefined time $T_{c2,i}$.

Through the above discussion, it is proven that the position $P_{i}$ of the quadrotor will track the planned position $\xi_{1,i}^{d}$ within a predetermined time.

#### 3.2.2. Predefined-Time Sliding Mode Controller Design of the Attitude Loop

This paper will prove that the attitude loop is stable in a predetermined time. Define the given input to the attitude loop as: $A_{i}^{d}={\left[\theta_{i}^{d},\phi_{i}^{d},\psi_{i}^{d}\right]}^{T}$. The attitude error is $E_{i,A}=A_{i}-A_{i}^{d}$.

In order to prove that the attitude loop error converges to 0 within the predefined time $T_{c4,i}$, the Lyapunov function is defined as:(46)$$V_{4,i}=\frac{1}{2}E_{i,A}^{T}E_{i,A}$$

The sliding mode switching surface is defined as:(47)$$S_{i,A}={\dot{E}}_{i,A}^{}+\Psi_{i,A}$$
(48)$$\Psi_{i,A}=\frac{\pi }{2\eta_{3,i}T_{c3,i}\sqrt{\alpha_{3,i}\beta_{3,i}}}\left(\alpha_{3,i}V_{4,i}^{-\eta_{3,i}/2}+\beta_{3,i}V_{4,i}^{\eta_{3,i}/2}\right)E_{i,A}$$

The parameters $\alpha_{3,i},\beta_{3,i},0<\eta_{3,i}<1$ are all positive numbers. When $S_{i,A}=0$, Equation (47) can be simplified as:(49)$${\dot{E}}_{i,A}=-\Psi_{i,A}$$

We take the derivative of Equation (46), and then substitute Equations (48) and (49) into the derivative of Equation (46):(50)$$\dot{V_{4,i}}=-\frac{\pi }{\eta_{3,i}T_{c3,i}\sqrt{\alpha_{3,i}\beta_{3,i}}}\left(\alpha_{3,i}V_{4,i}^{1-\eta_{3,i}/2}+\beta_{3,i}V_{4,i}^{1+\eta_{3,i}/2}\right)$$

By Lemma 1, $E_{i,A}$ will be stable for a predefined period of time $T_{c3,i}$. Next, this paper proves that the sliding mode surface $S_{i,A}$ is stable for a predefined time. Define the Lyapunov function:(51)$$V_{5,i}=\frac{1}{2}S_{i,A}^{T}S_{i,A}$$

By differentiating Equation (51), we can obtain:(52)$${\dot{V}}_{5,i}=S_{i,A}^{T}\left(F_{i}+\tau_{i}-\overset{¨}{A_{i}^{d}}+{\dot{\Psi }}_{i,A}\right)$$

Let the attitude loop control quantity be:(53)$$\tau_{i}=-F_{i}+\overset{¨}{A_{i}^{d}}-\Psi_{i,A}-\frac{\pi S_{i,A}}{2\left(\eta_{4,i}T_{c4,i}\sqrt{\alpha_{4,i}\beta_{4,i}}\right)}(\alpha_{4,i}V_{5,i}^{-\eta_{4,i}/2}+\beta_{4,i}V_{5,i}^{-\eta_{4,i}/2})$$

By substituting Equation (53) into Equation (52), we get:(54)$${\dot{V}}_{5,i}=-\frac{\pi }{\eta_{4,i}T_{c4,i}\sqrt{\alpha_{4,i}\beta_{4,i}}}\left(\alpha_{4,i}V_{4,i}^{1-\eta_{4,i}/2}+\beta_{4,i}V_{4,i}^{1+\eta_{4,i}/2}\right)$$

According to Lemma 1, $S_{i,A}$ will be stable for a predefined time $T_{c4,i}$.

Through the above discussion, this paper proves that the attitude $A_{i}$ of the quadrotor can track the attitude expectation indirectly calculated from the control quantity of the position loop within a predetermined time.

## 4. Simulation Results and Analysis

In this section, MATLAB/Simulink is used for numerical simulation to verify the correctness of the algorithm. The simulation verification in this section is mainly divided into three parts: 1. This paper compares the control effects of predefined-time sliding mode control and fixed-time sliding mode control on quadrotor formation; 2. This paper compares the control effects of predefined-time sliding mode control and adaptive predefined-time sliding mode control based on RBF neural networks on the quadrotor after adding interference into the position loop; 3. This paper verifies the effect of quadrotor formation tracking the expected formation and avoiding obstacles.

Consider the following quadrotor parameters: the set inertia torque $J=\left[\begin{array}{ccc} 1.22 & 0 & 0\\ 0 & 1.22 & 0\\ 0 & 0 & 1.22 \end{array}\right]\mathrm{kg}\cdot m^{2}$, gravitational acceleration $g=9.81\;{m/s}^{2}$, the distance of the motor from the quadrotor center of gravity l=0.21 m, the mass of the quadrotor m=1.1 kg, and the drag coefficient of the quadrotor in three axial directions: $k_{x}=0.1,k_{y}=0.1,k_{z}=0.1$.

### 4.1. Comparison of Predefined-Time Sliding Mode Control and Fixed-Time Sliding Mode Control

This section compares the difference between the predefined-time sliding mode control algorithm and the fixed-time sliding mode control algorithm [22] for the control effect of the quadrotor. The parameters of the predefined-time sliding mode control algorithm are described as follows: for the position loop, the predefined time of the position tracking error $E_{i,p}$ of the quadrotor is set as $T_{c1,i}=0.5$, the predefined time of the sliding mode surface is set as $T_{c2,i}=5$, and other parameters are set as $\alpha_{1,i}=2,\beta_{1,i}=$ $2,\alpha_{2,i}=1,\beta_{2,i}=2,\eta_{1,i}=0.35,\eta_{2,i}=0.35$; for the attitude loop, the predefined time of the attitude tracking error of the quadrotor is set as $T_{c3,i}=1$, the predefined time of the sliding mode switch function is set as $T_{c4,i}=0.5$, and the other parameters are set as $\alpha_{3,i}=50,\beta_{3,i}=150,\alpha_{4,i}=100,\beta_{4,i}=200,\eta_{3,i}=0.25,\eta_{4,i}=0.15$. The parameter setting of the fixed-time sliding mode control algorithm is the same as that in the literature [22].

Figure 4 compares the control effects of the predefined-time sliding mode control algorithm and the fixed-time sliding mode control algorithm on the tracking error of the quadrotor position loop. As can be seen from Figure 4, compared with the fixed-time sliding mode control, the convergence rate of the predefined-time sliding mode control is faster in the axis direction of x,y,z, and the position error curve of the quadrotor converges within 2 s under the predefined-time sliding mode control ($T_{c1,i}+T_{c2,i}=5.5$), which also accords with the control objective of the predefined time. The steady-state error of the predefined-time controller is obviously smaller than that of the fixed-time controller. Figure 5 compares the control effects of the predefined-time sliding mode control algorithm and the fixed-time sliding mode control algorithm on the tracking error of the quadrotor attitude loop. As can be seen from Figure 5, the stability of the attitude error curve of the predefined-time sliding mode control algorithm is worse than that of the fixed-time sliding mode control algorithm, but the convergence speed and the stability of the attitude error curve in the steady state of the predefined-time sliding mode control algorithm are similar to those of the fixed-time sliding mode control algorithm. In addition, the convergence time of the fixed-time sliding mode controller cannot be explicitly determined by parameters, but the convergence time of the predefined-time sliding mode controller can be explicitly determined by a certain parameter. To sum up, the predefined-time sliding mode controller is better than the fixed-time sliding mode controller. Both converge within 1 s ($T_{c3,i}+T_{c4,i}=1.5$), which also accords with the control objectives of the predefined-time controller, and the steady-state error of the predefined-time controller for attitude loop control is significantly smaller than that of the fixed-time controller. Figure 6 and Figure 7 are the control curves for lift and attitude torque of the quadrotor, respectively. We can see from the figure that the lift curve is relatively smooth and that no frequent shaking occurs. For the torque curve, there is no large shaking on the whole, but there will be a small amount of shaking.

### 4.2. Comparison of Predefined-Time Sliding Mode Control and Adaptive Predefined-Time Sliding Mode Control Based on an RBF Neural Network

This section mainly verifies the control effect difference between the adaptive predefined-time sliding mode control based on an RBF neural network and the predefined-time sliding mode control after adding position interference. The specific parameters are described as follows: The parameters of the predefined-time sliding mode control are exactly the same as those set in the previous section; the parameters of the adaptive predefined-time sliding mode control based on the RBF neural network, except for those of the RBF neural network, are the same as in the previous section. The parameters of the RBF neural network are set as follows: $I_{i}={\left[e_{i,z},{\dot{e}}_{i,z},s_{i,z}\right]}^{T}$,where $e_{i,z}$ represents the *Z*-axis control error, the width value $b_{i}$ of RBF neural network is set to 8, and the center value of the RBF neural network is set to $C_{i}=\left[C_{i,1},C_{i,2},C_{i,3},C_{i,4},C_{i,5}\right]$ $=\left[\begin{array}{ccccc} -6 & -3 & 0 & 3 & 6\\ -4 & -2 & 0 & 2 & 4\\ -8 & -4 & 0 & 4 & 8 \end{array}\right]$. Finally, the initial estimated value of the actual weight of the neural network is set as ${\hat{W}}_{i}={\left[w_{i,1},w_{i,2},w_{i,3},w_{i,4},w_{i,5}\right]}^{T}={\left[0,0,0,0,0\right]}^{T}$.

Figure 8 shows the comparison between the estimated value of unknown interference and the real value of the unknown interference of the adaptive predefined-time sliding mode control algorithm based on the RBF neural network. Since the actual unknown interference is not continuously excited, the estimated value of the neural network in Figure 8 can only be guaranteed to be bounded and cannot converge to the true value. Figure 9 shows the control effects of the above two control algorithms. As can be seen from Figure 10, after adding the model interference, the convergence rate of the adaptive predefined-time sliding mode controller based on an RBF neural network is faster than that of the predefined-time sliding mode controller. Since the reference trajectory is not continuously excited, the adaptive predefined-time sliding mode control based on the RBF neural network presented in this paper cannot completely eliminate the influence of interference. However, compared with the predefined-time sliding mode controller without RBF neural network, the adaptive predefined-time sliding mode controller based on RBF neural network can improve the convergence accuracy of the *Z*-axis error curve by 200 times and can also make the *Z*-axis position state stable in the predefined time $T_{c1,i}+T_{c2,i}=5.5$.

### 4.3. Simulation Results of the Obstacle Avoidance of the Quadrotor Formation

In this section, the simulation shows that the position and attitude of the quadrotor can be controlled by the adaptive predefined-time sliding mode control algorithm based on the RBF neural network. On this basis, we use the artificial potential field method to achieve obstacle avoidance in formation. In addition, this paper also optimizes the artificial potential field method, adding disturbance to avoid falling into the local optimal. The formation mainly consists of three quadrotors (one leader and two followers). The expected pilot trajectory is set as follows: $\xi_{1,1}^{r}={\left[0,t,t\right]}^{T}$, the trajectory is moving uniformly with a velocity of 1 m/s in the y and z directions), The trajectories of the first follower and the second follower are, respectively, $\xi_{2,1}^{r}=\xi_{1,1}^{r}+{\left[60,0,0\right]}^{T}$ and $\xi_{3,1}^{r}=\xi_{1,1}^{r}-{\left[60,0,0\right]}^{T}$, and the coordinates of obstacle 1, obstacle 2, and obstacle 3 are, respectively, $O_{1}=\left[0,80,80\right]m^{T}$,$O_{2}=\left[60,120,120\right]m^{T}$, $O_{3}=\left[-60,120,120\right]m^{T}$. The radius of the artificial potential field is $R_{\varsigma }=5$, while the collision avoidance radius between the formation and obstacles is $R_{k}=50$.

Figure 9 shows the comparison between the planned trajectory and the real trajectory of the quadrotor formation with obstacles. According to that, we can see the planned trajectory can be prevented from falling into local optimal when encountering obstacles by adding virtual force to the artificial potential field method, which can help the formation successfully bypass the obstacles and finally return to the planned path. At the same time, each quadrotor can stably track the planned trajectory under the control of the adaptive predefined-time sliding mode control algorithm based on the RBF neural network, able to stably track the planned trajectory. Taking the leader as an example, this paper shows the error curve of the leader tracking the planned trajectory on the axes in Figure 11, Figure 12 and Figure 13. It can be seen that the quadrotor can track the planned trajectory stably and quickly under the control of the adaptive pre-defined time sliding mode control algorithm based on the RBF neural network, and the maximum tracking error in the obstacle avoidance process will not exceed 0.05 m.

## 5. Conclusions

To solve the obstacle avoidance problem of a quadrotor formation model with unknown interference, this paper proposed an artificial potential field method for path planning. It adopted the adaptive predefined-time sliding mode controller based on the RBF neural network to control the formation. Firstly, we converted the ideal trajectory $\xi_{i,1}^{r}$ into the actual planned trajectory $\xi_{i,1}^{d}$ using the artificial potential field method and input the actual planned trajectory $\xi_{i,1}^{d}$ to the position loop controller of each quadrotor in the formation. Secondly, the RBF neural network was used to compensate for the unknown disturbance of the Z-axis of the quadrotor, and the predefined-time sliding mode control algorithm was used to control the position of the quadrotor. Thirdly, we transformed the lift force $T_{i}$ of the position loop into the expected value of the attitude loop through Equations (11) and (12) and input that to the attitude loop controller. Finally, we could realize the position and attitude control of the quadrotor, then it could track the planned trajectory $\xi_{i,1}^{d}$ within a predefined time, and finally we could realize the trajectory control of the whole formation. In future work, we will verify the algorithm’s accuracy and correct its shortcomings by combining the real quadrotor formation.

## Figures and Tables

**Figure 1 sensors-23-02392-f001:**
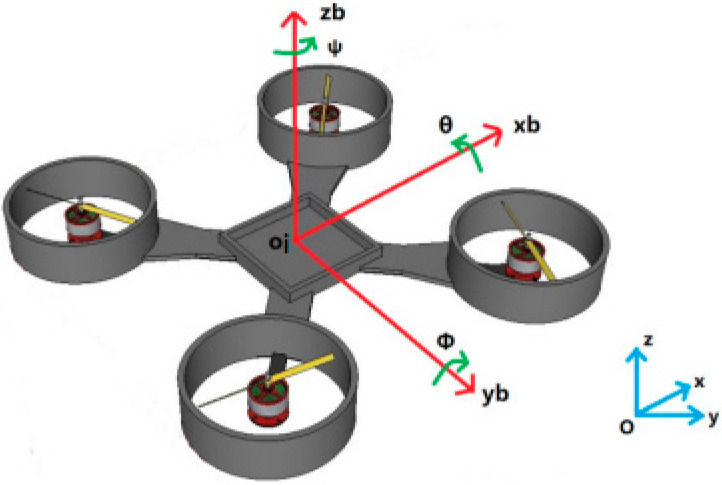
Quadrotor model diagram.

**Figure 2 sensors-23-02392-f002:**
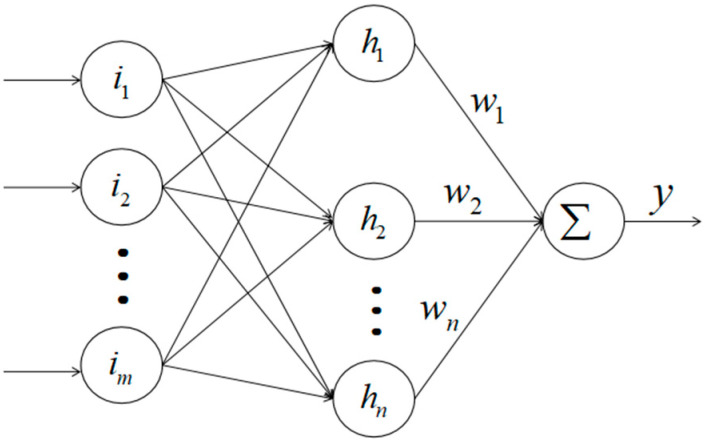
RBF neural network.

**Figure 3 sensors-23-02392-f003:**
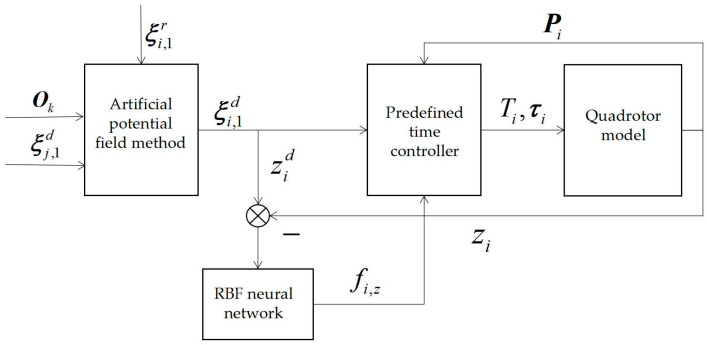
Overall design flow chart.

**Figure 4 sensors-23-02392-f004:**
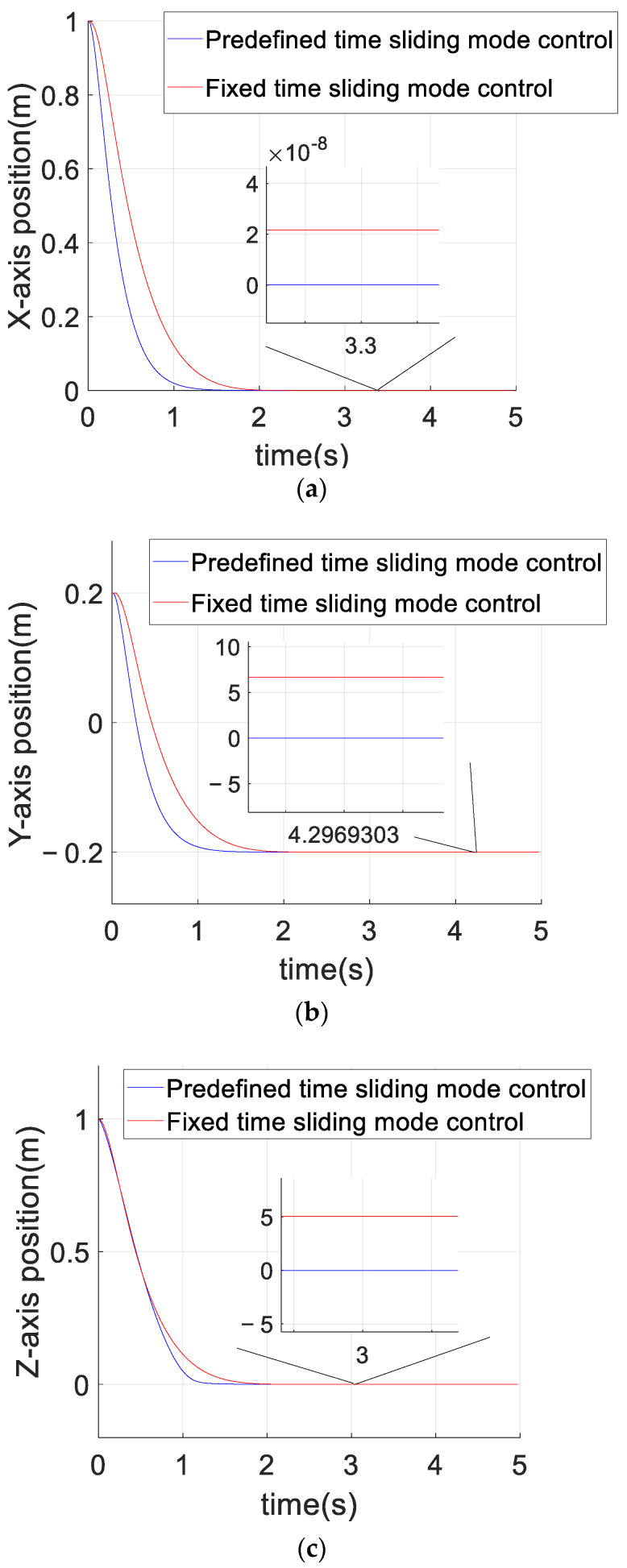
Comparison of position error between the predefined-time sliding mode control and the fixed-time sliding mode control. (**a**) *X*-axis position error curve. (**b**) *Y*-axis position error curve. (**c**) *Z*-axis position error curve.

**Figure 5 sensors-23-02392-f005:**
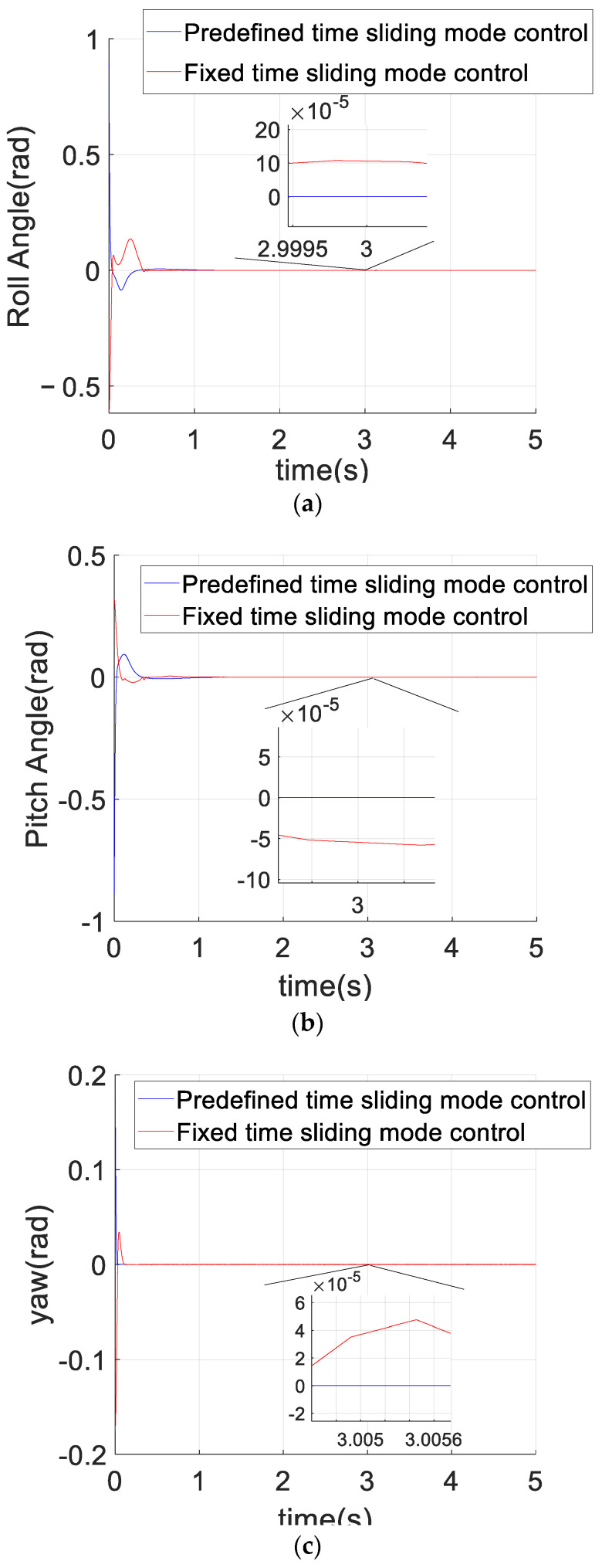
Attitude error comparison between the predefined-time sliding mode control and the fixed-time sliding mode control. (**a**) Roll-Angle error curve. (**b**) Pitch-Angle error curve. (**c**) Yaw-Angle error curve.

**Figure 6 sensors-23-02392-f006:**
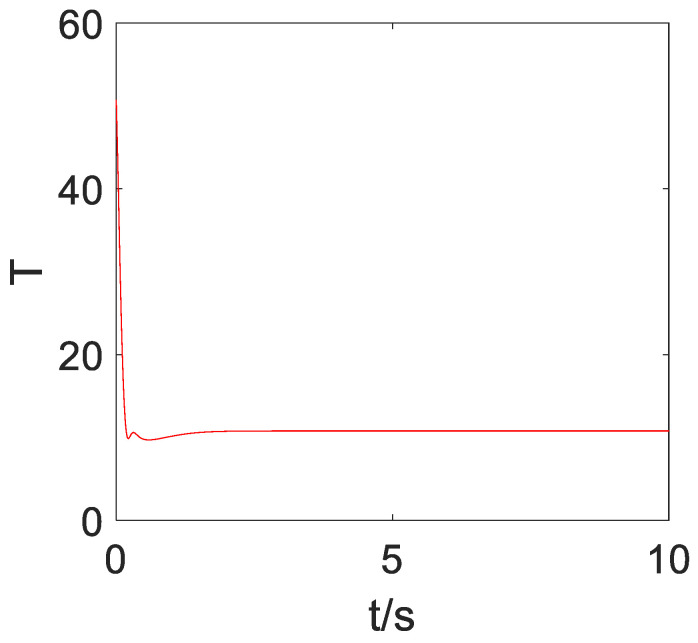
Lift control curve.

**Figure 7 sensors-23-02392-f007:**
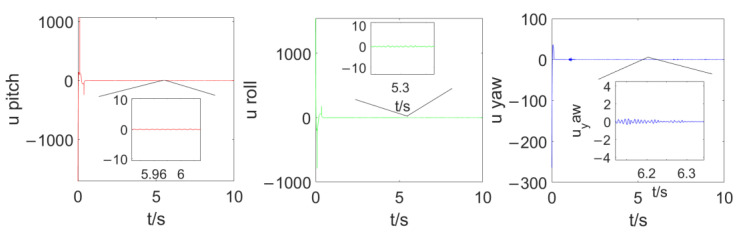
Attitude torque curve.

**Figure 8 sensors-23-02392-f008:**
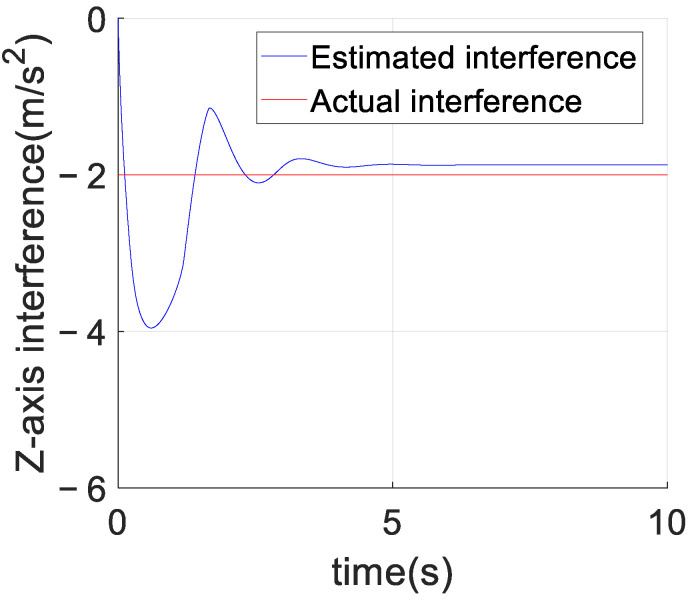
Comparison between estimated disturbance and actual disturbance of the adaptive predefined-time sliding mode control based on the RBF neural network.

**Figure 9 sensors-23-02392-f009:**
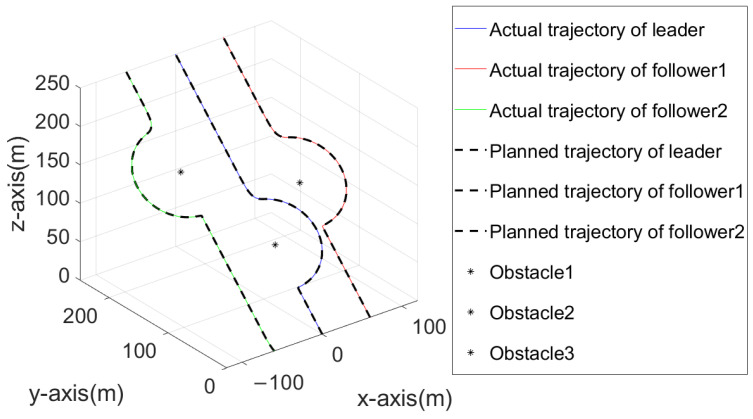
Three-dimensional comparison diagram of planned trajectory and actual trajectory of quadrotor formation.

**Figure 10 sensors-23-02392-f010:**
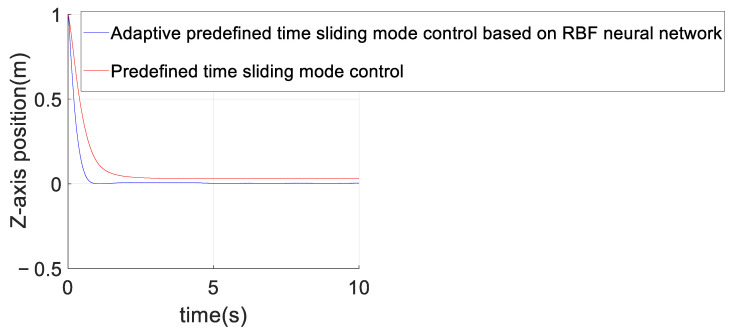
Comparison of *Z*-axis position error between the adaptive predefined-time sliding mode control based on the RBF neural network and the predefined-time sliding mode control.

**Figure 11 sensors-23-02392-f011:**
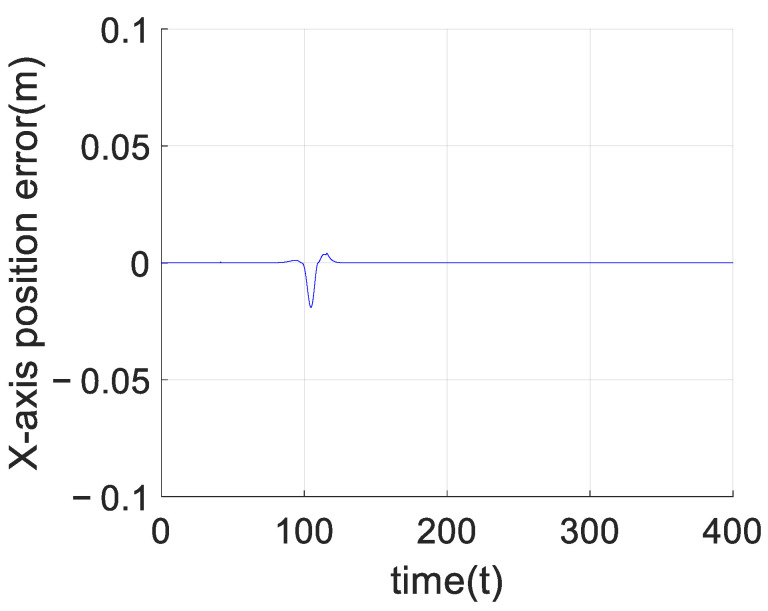
*X*-axis tracking error of the leader.

**Figure 12 sensors-23-02392-f012:**
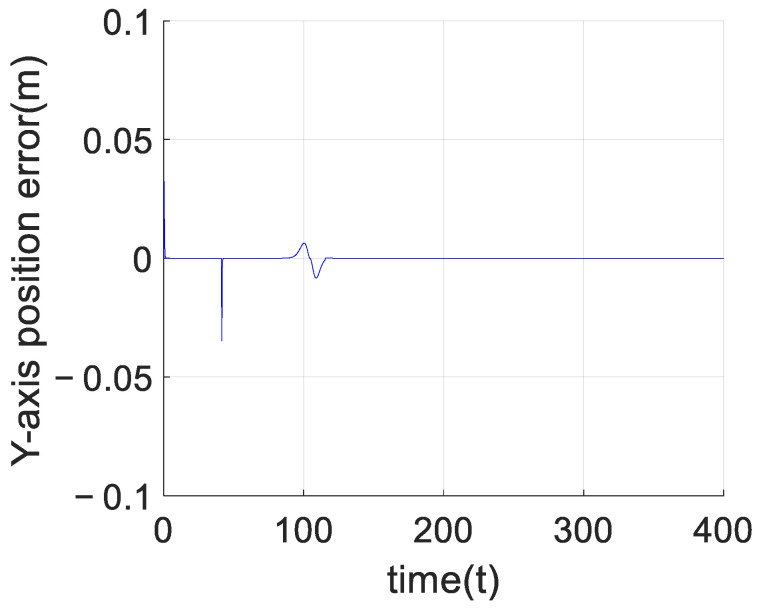
*Y*-axis tracking error of the leader.

**Figure 13 sensors-23-02392-f013:**
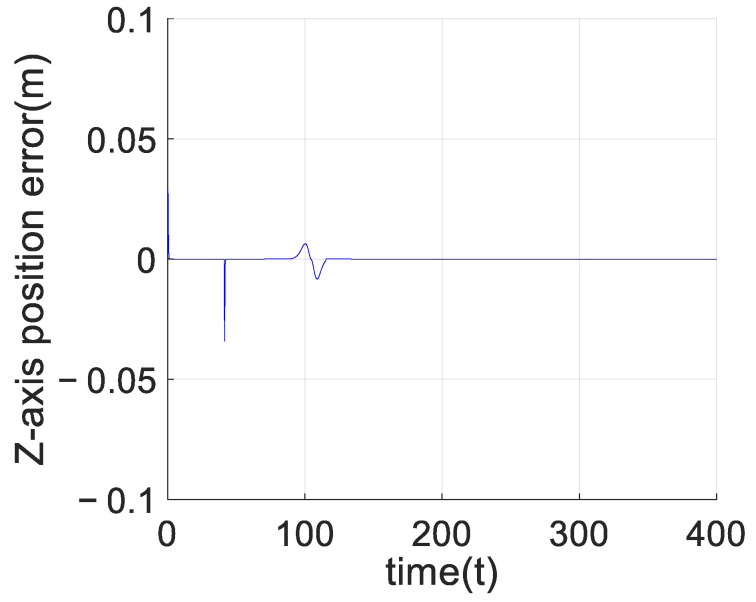
*Z*-axis tracking error of the leader.

## Data Availability

Not applicable.

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
