# Peer review of "Adaptive Predefined-Time Sliding Mode Control for QUADROTOR Formation with Obstacle and Inter-Quadrotor Avoidance"

_sensors, 2023, doi:10.3390/s23052392_

Round 1

Reviewer 1 Report

Upon review it is not clear that this submission is appropriate for Sensors; Drones or Robotics may be a better selection.

The paper needs some English-language editing with an English language technical reviewer.

It is stated without evidence or references that RBFNN are an effective tool for estimation of unknown interference (line 187); can a reference be provided?

The coordinate system description in section 2.2 (lines 136+) is difficult to follow; would subscripts make this easier?

The description of the RBFNN has some language which is confusing.  In particular what is the center and width of the neural network?  I believe this refers to the mean and variance of the basis functions but the language is not clear.

The plots are a little confusing; I understand the inserts (such as in figure 3, around the time = 3s) are used to 'zoom in' to the curves but can this be represented in a more clear fashion or in a tabular / final error statement?

Reviewer 2 Report

The authors propose the use of artificial potential field method with virtual force to plan the obstacle avoidance path of quadrotor formation to solve the problem that artificial potential field method may get trapped in a local optimal solution.  The authors claim that this work makes the planned trajectory of the quadrotor formation avoid obstacles and leads to the convergence of the error between the true trajectory and the planned trajectory within a predetermined time under the premise of adaptive estimation of unknown interference in the quadrotor model.

This paper suggests an artificial potential field method with virtual force for trajectory planning of quadrotor formation along with an adaptive predefined time sliding mode control algorithm based on RBF neural network to perform the position and attitude control of the quadrotor. The paper mainly solves the issue of the quadrotor control algorithm with fixed time whose convergence time is lousy. Also, it combines the predefined time algorithm with adaptive algorithms represented by the RBF neural network.

The paper mainly proposes an artificial potential field method with virtual force o solve the local optimal trapping problem when using artificial potential field method. Also, a preset time sliding mode control algorithm for controlling the position is contrasted to the fixed-time sliding mode approach in terms of convergence time. Finally, an adaptive predefined time sliding mode control algorithm based on RBF neural network is proposed to tackle the occasions where there is interference in the environment or inaccurate modeling of the quadrotor model.

The procedure followed by the authors is clear and the paper does not require any further improvements in terms of methodology.

The conclusions well summarize the work results and highlight the key findings.

The list of references is adequate and completely summarizes the state of art findings on the topic of the paper.

Reviewer 3 Report

The manuscript investigates a very interesting topic on the adaptive predefined time sliding mode control for quadrotor formation. The research structure of the manuscript is clearly organized.

Please add a graphic abstract at the end of the Introduction

Please enlarge all the figures with high resolutions.

Please improve the description in the conclusions according to the achievements in this study.

I encourage the authors to double-check the spelling in the text and continue to perform a further investigation on this study.

Reviewer 4 Report

In this paper, Liu et al. proposed a predefined time sliding model control algorithm built upon RBF neural network for control and obstacle avoidance of quadrotor formation. This work has clear approach and proof with great presentation of ideas and results. It’s in a publishable format, only a few comments added below:

1.     The authors wrote: “In order to present the artificial potential field method between…”. 
May the authors explain the selection of angle threshold, and expected results if not with such constraint?

2.     In Figure 4(b), the Pitch Angle error curve showed unstable performance of Predefined time sliding model control in the early time compared to Fixed time sliding mode control. May the authors explain your understanding of why such instability happened and why Fixed time sliding mode control is not preferred given this figure. 

Round 2

Reviewer 1 Report

Another quick-look at the English usage